# In Vitro and In Silico Analysis of PTP1B Inhibitors from *Cleistocalyx operculatus* Leaves and Their Effect on Glucose Uptake

**DOI:** 10.3390/nu16172839

**Published:** 2024-08-24

**Authors:** Jorge-Eduardo Ponce-Zea, Byeol Ryu, Ju-Yong Lee, Eun-Jin Park, Van-Hieu Mai, Thi-Phuong Doan, Hee-Ju Lee, Won-Keun Oh

**Affiliations:** 1Research Institute of Pharmaceutical Sciences, College of Pharmacy, Seoul National University, Seoul 08826, Republic of Korea; jepz210689@snu.ac.kr (J.-E.P.-Z.); estrella56@snu.ac.kr (B.R.); sbplee@snu.ac.kr (J.-Y.L.); eunjin_p@snu.ac.kr (E.-J.P.); maihieu@snu.ac.kr (V.-H.M.); phuongdoan@snu.ac.kr (T.-P.D.); 2Natural Product Informatics Research Center, Korea Institute of Science and Technology, Gangneung 25451, Republic of Korea; hjlee81@kist.re.kr

**Keywords:** *Cleistocalyx operculatus*, Myrtaceae, protein tyrosine phosphatase 1B (PTP1B), kinetic, molecular docking, glucose uptake

## Abstract

As part of our ongoing research on new anti-diabetic compounds from ethnopharmacologically consumed plants, two previously undescribed lupane-type triterpenoids (**1** and **2**) with dicarboxylic groups, an undescribed nor-taraxastane-type triterpenoid (**3**), and 14 known compounds (**4**–**17**) were isolated from the leaves of *Cleistocalyx operculatus*. Extensive spectroscopic analysis (IR, HRESIMS, 1D, and 2D NMR) was used for structure elucidation, while the known compounds were compared to reference data reported in the scientific literature. All the isolates (**1**–**17**) were evaluated for their inhibitory effects on the protein tyrosine phosphatase 1B (PTP1B) enzyme. Compounds **6**, **9**, and **17** showed strong PTP1B inhibitory activities. The mechanism of PTP1B inhibition was studied through enzyme kinetic experiments. A non-competitive mechanism of inhibition was determined using Lineweaver–Burk plots for compounds **6**, **9**, and **17**. Additionally, Dixon plots were employed to determine the inhibition constant. Further insights were gained through a structure–activity relationship study and molecular docking analysis of isolated compounds with the PTP1B crystal structure. Moreover, all isolates (**1**–**17**) were tested for their stimulatory effects on the uptake of 2-deoxy-2-[(7-nitro-2,1,3-benzoxadiazol-4-yl) amino]-D-glucose (2-NBDG) in differentiated 3T3-L1 adipocyte cells. Compounds **6**, **13**, and **17** exhibited strong glucose absorption stimulation activity in a dose-dependent manner.

## 1. Introduction

The prevalence of diabetes has been steadily increasing globally in recent decades, with an estimated 10.5% of individuals aged between 20 and 79 diagnosed with the disease in 2021 [1]. Within this age range, diabetes is a risk factor in one out of every nine deaths resulting from complications associated with the condition, underscoring its significance as a major public health concern worldwide [2,3]. Despite the development of numerous non-insulin glucose-lowering agents for managing diabetes, controlling target blood glucose levels remains challenging and often results in undesirable side effects [4,5]. Hence, there is an ongoing need to investigate new substances to prevent or treat diabetes more effectively. Insulin resistance is a central factor in the pathogenesis of type 2 diabetes [6]. In the manifestation of insulin resistance, protein tyrosine phosphatase 1B (PTP1B) plays a significant role as a negative regulator of insulin signaling [7,8,9,10,11]. Thus, PTP1B has been identified as a promising target for developing improved glucose-regulating agents [12].

Medicinal plants have been used empirically and reported to be helpful in diabetes treatment [13,14]. The mechanisms of action attributed to reported anti-diabetic plants include insulin mimetics, increased insulin secretion, β-cells regeneration in the pancreas, reduced glucose absorption, reduced glucosidase and amylase activity, and radical scavenging activity [13,15]. Additionally, it has been observed that some plant extracts and their constituents exhibit PTP1B inhibitory action [16]. However, despite many research efforts, no approved medication currently targets PTP1B inhibition. *Cleistocalyx operculatus* (Roxb. Merry. Et Perry) is an evergreen plant that has been historically utilized in Vietnam and China as traditional herbal tea and medicine to treat various ailments such as colds, fever, dysentery, vomiting, indigestion, gastrointestinal disorders, and inflammation [17,18,19,20]. Interestingly, an aqueous extract from flower buds of *C. operculatus* was reported to decrease glucose levels in individuals who were administered the extract for 12 weeks [21]. Chalcone-meroterpene derivatives isolated from the buds of *C. operculatus* have shown PTP1B inhibition activity [22]. Other chemical constituents identified in the plant include flavanones, flavones, and oleanane- and ursane-type triterpenoids [23,24,25].

Compared to the extensive research conducted on the flower buds of *C. operculatus*, there have been relatively fewer studies on its leaves. Thus, our research aimed to identify potential compounds with anti-diabetic properties in the more readily available leaves of *C. operculatus*. This led to the isolation of 17 compounds belonging to different classes of triterpenoids, including lupane, taraxastane, ursane, and oleanane type. Among these, two undescribed lupane-type triterpenoids (**1** and **2**) with dicarboxylic groups and one undescribed nor-taraxastane-type triterpenoid (**3**), were isolated. All the isolates (**1**–**17**) were evaluated for their inhibitory effects on PTP1B and also their stimulatory effects on 2-NBDG uptake in 3T3-L1 adipocyte cells. We elucidated the inhibition mechanisms on PTP1B and analyzed the structure–activity relationships. Furthermore, we investigated whether the PTP1B inhibitory activity correlated with glucose uptake stimulation in differentiated 3T3-L1 adipocyte cells.

## 2. Materials and Methods

### 2.1. General Experimental Procedures

A JASCO P-2000 polarimeter (JASCO International Co., Ltd., Tokyo, Japan) with a 1 cm microcell was employed to measure optical rotations. UV spectra were recorded on a SpectraMax M5 (Molecular Devices, Sunnyvale, CA, USA), and IR spectra were obtained in a JASCO FT/IR-4200 spectrometer (JASCO International Co., Ltd., Tokyo, Japan). Waters Xevo G2 QTOF mass spectrometer (Waters MS Technologies, Manchester, UK) equipped with an electrospray interface (ESI) was used to obtain high-resolution mass spectrometry data. NMR spectra were acquired on a JEOL 600 MHz NMR spectrometer (Jeol Ltd., Tokyo, Japan). Column chromatography resins employed in isolation included reversed-phase silica gel (ODS-A 12 nm S-150 μm, YMC Co, Ltd., Kyoto, Japan), Diaion HP-20 (Mitsubishi Chemical Corp., Tokyo, Japan), and Sephadex LH-20 (GE Healthcare, Little Chalfont, UK). Medium-pressure chromatographic separations employed the MPLC-Reveleris system (Grace, Columbia, MD, USA) with a Reveleris flash cartridge column (C18, 120 g, Grace, Columbia, MD, USA). Semi-preparative HPLC system included a Gilson 321 pump (Gilson Inc., Middleton, WI, USA) and a Gilson UV/VIS-151 detector (Gilson Inc., Middleton, WI, USA), and a YMC-Triart C18 column (250 × 10 mm ID, 5 μm, YMC Co., Ltd., Kyoto, Japan). Solvents were purchased from Daejung Chemicals & Metals Co. Ltd., Siheung, Republic of Korea.

### 2.2. Plant Material

The leaves of *Cleistocalyx operculatus* were purchased from a market in My Duc district, Hanoi, Vietnam, in February 2019. The botanical identification of the specimen was performed by Prof. Won Keun Oh. A voucher specimen (SNU2019-02) was deposited at the Laboratory of Pharmacognosy, College of Pharmacy, Seoul National University, Seoul, Republic of Korea.

### 2.3. Extraction and Isolation

Air dried leaves of *C*. *operculatus* (2.5 kg) were powdered and extracted three times with 50% aqueous EtOH at 60 ℃ for 8 h each. The crude extract obtained (427.8 g) was suspended in water and applied to a Diaion HP-20 column, then eluted sequentially with 50%, 70%, and 100% EtOH, followed by acetone to yield four fractions (A–D). Fraction C was further fractionated in a silica gel column chromatography (silica gel, 60–200 mesh; Zeochem, Louisville, KY, USA), where *n*-hexane/EtOAc [1:0, 1:1, 1:9, MeOH; *v*/*v*] were used as solvent system, yielding subfractions C1–C9. Subfraction C4 was subjected HPLC [Optima Pak C18 column (10 × 250 mm, 5 μm particle size, R.S. Tech, Cheongju-si, Republic of Korea); mobile phase CH_3_CN in H_2_O containing 0.1% formic acid (0−43 min: 85−95% CH_3_CN, 43−50 min: 100% CH_3_CN); flow rate 2 mL/min] to yield compounds **1** (3.4 mg), **5** (5.0 mg), **9** (13.0 mg), and **12** (14.9 mg). Subfraction C7 was likewise fractionated using an open column RP-C18 (4 × 40 cm; 75 μm particle size) and eluted with a stepwise gradient of CH_3_CN/H_2_O (1:1 to 1:0) to afford nine subfractions (C7-1−C7-9). One of these subfractions, C7-4, was applied to Sephadex LH-20 with H_2_O/MeOH (1:1 to 0:1) to obtain three subfractions (C7-4-1–C7-4-3). Subfraction C7-4-2 was subjected to RP-C18 column chromatography with (4 × 40 cm; 75 μm particle size) to yield five fractions (C7-4-2-1–C7-4-2-5). Subfraction C-7-4-2-2 was directly purified by HPLC [Optima Pak C18 Column; 10 × 250 mm, 5 μm particle size, R.S. Tech, Cheongju-si, Republic of Korea; mobile phase CH_3_CN in H_2_O containing 0.1% formic acid (0−30 min: 52% CH_3_CN, 43−52 min: 100% CH_3_CN); flow rate 2 mL/min], producing **7** (5.5 mg). Subfraction C7-4-2-4 was further separated by HPLC [Optima Pak C18 column (10 × 250 mm, 5 μm particle size, R.S. Tech, Cheongju-si, Republic of Korea); mobile phase CH_3_CN in H_2_O containing 0.1% formic acid (0−43 min: 52% CH_3_CN, 43−52 min: 100% CH_3_CN); flow rate 2 mL/min], yielding compounds **3** (6.4 mg) and **14** (20.0 mg). Subfraction C7-7 was applied to Sephadex LH-20 with H_2_O/MeOH (1:1 to 0:1) to afford seven subfractions (C7-7-1–C7-7-7). Among them, subfraction C7-7-4 was subjected to RP-C18 (4 × 40 cm; 75 μm particle size) column chromatography to yield eight subfractions using a gradient from 50% to 100% CH_3_CN. Subfraction C7-7-4-1 was directly purified by HPLC [Optima Pak C18 column (4.6 × 250 mm, 5 μm particle size, R.S. Tech, Cheongju-si, Republic of Korea); mobile phase CH_3_CN in H_2_O containing 0.1% formic acid (0−30 min: 36–51% CH_3_CN, 33–40 min: 100% CH_3_CN); flow rate 1 mL/min], resulting in compound **10** (4.4 mg). Subfraction C8 was applied to Sephadex LH-20 with CH_2_Cl_2_/MeOH (1:1 to 0:1) to afford six fractions (C8-1–C8-6). Subfraction C8-4 was subjected to RP-C18 column chromatography (4 × 40 cm; 75 μm particle size) and eluted with a stepwise gradient of CH_3_CN/H_2_O (1:1 to 1:0) to afford seven subfractions (C8-4-1–C8-4-7). Subfraction C8-4-6 was then applied to RP-C18, yielding seven fractions (C8-4-6-1–C8-4-6-7). Subfraction C8-4-6-5 was directly purified by HPLC [Optima Pak C18 Column; 10 × 250 mm, 5 μm particle size, R.S. Tech, Cheongju-si, Republic of Korea; mobile phase CH_3_CN in H_2_O containing 0.1% formic acid (0−40 min: 50–75% CH_3_CN, 41–50 min: 100% CH_3_CN); flow rate 2 mL/min], producing compounds **2** (7.9 mg), **11** (4.5 mg), **13** (11.6 mg), and **17** (3.4 mg). Subfraction C8-4-6-6 was purified using the same system [Optima Pak C18 Column; 10 × 250 mm, 5 μm particle size, R.S. Tech, Cheongju-si, Republic of Korea; mobile phase CH_3_CN in H_2_O containing 0.1% HCO_2_H (0−40 min: 46% CH_3_CN, 41–50 min: 100% CH_3_CN); flow rate 2 mL/min], resulting in compound **16** (4.1 mg).

### 2.4. Spectroscopic and Physical Characteristic of Compounds

Cleistocalyxic acid L (**1**): brownish gum; ${\left[\alpha \right]}_{D}^{25}$: −21 (c 0.3, MeOH); UV λmax (MeOH) (log ɛ) (nm) 224 (1.96); IR (KBr) νmax 3364, 2946, 2871, 1706, 1697, 1451, 1389, 1219, and 1032 cm^−1^; HR-ESI-MS *m*/*z* 515.3382 [M − H]^−^ (calcd for C_31_H_47_O_6_, 515.3373); ^1^H and ^13^C NMR data (pyridine-*d*5, 600 and 150 MHz); see Table 1. The original IR, (−)-HR-ESI-MS, ^1^H NMR, ^13^C NMR, HSQC, HMBC, and NOESY spectra are shown in Appendix A.

Cleistocalyxic acid M (**2**): white amorphous powder; ${\left[\alpha \right]}_{D}^{25}$: +4 (c 0.3, MeOH); UV λ_max_ (MeOH) (log ɛ) (nm) 226 (2.28). IR (KBr) ν_max_ 3245, 2934, 2889, 2359, 2342, 1718, 1680, 1641, 1452, 1212, 1167, and 1033 cm^−1^; HRESIMS *m*/*z* 515.3372 [M − H]^−^ (calcd for C_31_H_47_O_6_, 515.3373); ^1^H and ^13^C NMR data (pyridine-*d*_5_, 600 and 150 MHz); see Table 1. The original IR, (−)-HR-ESI-MS, ^1^H NMR, ^13^C NMR, HSQC, HMBC, and NOESY spectra are shown in Appendix A.

Cleistocalyxolide C (**3**): white amorphous powder; ${\left[\alpha \right]}_{D}^{25}$: +82 (c 0.3, MeOH). UV λ_max_ (MeOH) (log ɛ) (nm) 200 (3.16); IR (KBr) ν_max_ 3365, 2924, 1775, 1647, 1356, 1218, 1149, 1057 and 1033 cm^−1^; HRESIMS *m*/*z* 513.2851 [M + HCOO]^−^ (calcd for C_30_H_41_O_7_, 513.2852); ^1^H and ^13^C NMR data (pyridine-*d*_5_, 600, and 150 MHz); see Table 1. The original IR, (−)-HR-ESI-MS, ^1^H NMR, ^13^C NMR, HSQC, HMBC, and NOESY spectra are shown in Appendix A.

### 2.5. In Vitro PTP1B Inhibition Assay

PTP1B enzyme inhibition activity was assessed following a previously described protocol [26]. Briefly, 50 μL of 4 mM *p*-NPP was added to a buffer solution containing 1 mM dithiothreitol (DTT), 0.1 M NaCl, 1 mM EDTA, 50 mM citrate (pH 6.0), and 16 nM PTP1B (Enzo Life Sciences Inc., Farmingdale, NY, USA), along with either the test compounds or dimethyl sulfoxide (DMSO), bringing the final volume to 100 μL in a 96-well half-volume plate. It was followed by incubation at 37 °C for 30 min. After that, 10 μL of 10 M NaOH was added to quench the reaction. The enzymatic reaction product was measured by measuring the absorbance in a UV plate reader () at 405 nm. Ursolic acid was utilized as a positive control. The non-enzymatic reaction of the substrate was considered and accounted for by comparing data to a control experiment without the PTP1B enzyme. IC_50_ values were calculated using non-linear regression in GraphPad Prism 10 (GraphPad Software, Inc., San Diego, CA, USA), based on triplicate experiments.

### 2.6. Kinetic Analysis with PTP1B

Reaction velocity was measured at different concentrations of tested compounds for kinetic analysis. The tested compounds or DMSO were preincubated with the enzyme and buffer solution at 37 °C for 2 min, followed by adding 50 μL of substrate (*p*-NPP) at different concentrations. The initial reaction velocities were monitored at 37 °C at 405 nm for 15 min. The kinetic reaction slopes were calculated based on the absorbance increments observed between 2 and 10 min (within the linear range). GraphPad Prism 10 (GraphPad Software, Inc., San Diego, CA, USA) was used to graph the double reciprocal Lineweaver–Burk and Dixon plots.

### 2.7. Cell Culture and Differentiation of 3T3-L1 Adipocytes

3T3-L1 fibroblasts were cultured in DMEM (HyClone, Logan, UT, USA) supplemented with 10% calf serum, 100 U/mL penicillin, and 100 μg/mL streptomycin from Gibco (Grand Island, NY, USA). The cells were incubated at 37 °C with 5% CO_2_. Cells were differentiated after treatment with DMEM enriched with 10% fetal bovine serum (FBS) (HyClone, Logan, UT, USA), 1 μM dexamethasone (Sigma, St. Louis, MO, USA), 520 μM 3-isobutyl-1-methylxanthine (Sigma, St. Louis, MO, USA), and 1 μg/mL insulin (Roche, Mannheim, Germany). After 48 h, the media were changed with fresh DMEM containing 10% FBS, 1 μg/mL insulin, 100 U/mL penicillin, and 100 μg/mL streptomycin. Every 48 ours, the media were replaced until the induction of adipogenesis. Intracellular lipid droplets appeared after 4 to 6 days of incubation.

### 2.8. Measurement of Glucose Uptake Using the 2-NBDG Probe

In vitro experiments measured glucose uptake in 3T3-L1 adipocytes using 2-NBDG, a fluorescent glucose derivative (Invitrogen, Eugene, OR, USA). Cells were cultured on 96-well plates in a glucose-free medium supplemented with 10% FBS. After incubation, cells were treated with either insulin (as a positive control) or the test compounds in the presence or absence of 2-NBDG and were further incubated for an additional hour. Cells were washed with cold phosphate-buffered saline (PBS). The 2-NBDG fluorescence was quantified by measuring the signal intensity at excitation/emission wavelengths of 450/535 nm using a SpectraMax Gemini XPS fluorescence microplate reader (Molecular Devices, San Jose, CA, USA). For cellular transport studies involving 2-NBDG, 3T3-L1 adipocytes were grown on sterilized glass coverslips in a glucose-free medium containing 10% FBS for 24 h. After treatment with 2-NBDG, cells were washed with cold PBS. Fluorescent images were obtained with an Olympus IX 70 fluorescence microscope (Olympus Corporation, Tokyo, Japan) to quantify the cellular transport of 2-NBDG.

### 2.9. Molecular Docking Studies on the PTP1B Enzyme

The chemical structures of compounds **1**–**17** intended for docking simulations were minimized using Chem3D (Perkin Elmer, Shelton, CT, USA) and saved in .mol format. These files were then imported into Discovery Studio (Dassault Systèmes Biovia Corp., Vélizy-Villacoublay, France), where various conformers were generated, and ionization states were produced using the ‘Prepare Ligands’ tool. Energy minimization was performed by applying CHARMm force field and the Momany–Rone method. The structural data of the PTP1B protein were acquired from the RCSB Protein Data Bank, referencing PDB:1T49 [27]. For the preparation of the protein structure, parameters were set to CHARMm minimization, with protonation adjusted to pH 7.4, an ionic strength of 0.145, and the removal of water molecules. The docking procedure was conducted employing the Libdock protocol [28]. The coordinates set for identifying the sphere of protein–ligand interactions within the allosteric binding site of PTP1B were as follows: 53.8545, 30.4053, 24.5127, and 8.9. Default parameters were selected for docking preferences, setting the quality to high, choosing the FAST method for conformation, and applying the Steepest Descent for the Minimization Algorithm. Finally, the Discovery Studio Visualizer (Dassault Systèmes Biovia Corp., Vélizy-Villacoublay, France) was used to display the binding poses, protein surfaces, and interactions between proteins and ligands.

### 2.10. Statistical Analysis

Data were processed through variance analysis (ANOVA) to determine the significance of differences between groups, followed by Tukey’s or Duncan’s post hoc test. A *p*-value < 0.05 was considered indicative of a significant difference, with levels of significance denoted as follows: * *p* < 0.05, ** *p* < 0.01, and *** *p* < 0.001.

## 3. Results

### 3.1. Structure Elucidation of Compounds ***1**−**3*** from Cleistocalyx Operculatus

A 50% EtOH extract of the dried leaves of *C. operculatus* was separated using a Diaion^®^ HP-20 column with EtOH/H_2_O gradients, yielding four fractions (50%, 70%, 100%, and acetone). From the 100% EtOH fraction, three previously undescribed triterpenes (**1**–**3**) and 14 known compounds (**4**–**17**) were isolated by RP-18 and preparative high-performance liquid chromatography (HPLC) (Figure 1).

Compound **1** was obtained as a white amorphous powder with an optical rotation of ${\left[\alpha \right]}_{D}^{20}$ −21 (*c* 0.3, MeOH). High-resolution electrospray ionization mass spectrometry (HRESIMS) exhibited an ion peak at *m*/*z* 515.3382 [M − H]^−^ (calcd for C_31_H_47_O_6_, 515.3373), suggesting a molecular formula of C_31_H_48_O_6_. The ^1^H NMR spectrum of compound **1** (Table 1) showed four tertiary methyl groups (*δ*_H_ 2.28, 1.29, 1.20, and 1.20, each 3H, s), a methoxy group (*δ*_H_ 3.77, 3H, s), a hydroxyl methylene group (*δ*_H_ 4.73 and 4.56, each 1H, d, *J* = 15.0 Hz), an oxygenated methine proton (*δ*_H_ 3.27, 1H, dd, *J* = 11.4, 4.8 Hz), and two olefin protons (*δ*_H_ 5.61 and 5.38, each 1H, br s). The ^13^C NMR spectrum (Table 1) exhibited 31 carbon resonances, including two carboxylic groups at *δ*_C_ 178.8 and 177.3, an exocyclic methylene group at *δ*_C_ 107.3 and 156.5, an oxygenated carbon at *δ*_C_ 78.3, a methoxy group at *δ*_C_ 51.9, a hydroxyl methylene carbon at *δ*_C_ 64.7, and four methyl groups (*δ*_C_ 29.0, 17.9, 17.5, and 17.0). The ^1^H and ^13^C NMR spectroscopic data of compound **1** were found to be similar to those of melaleucic 28-*O*-methyl ester [29], except for a difference at C-30. In compound **1**, the C-30 methyl present in melaleucic 28-*O*-methyl ester was replaced by a hydroxyl methylene group at *δ*_C_ 64.7. A detailed analysis involving HSQC and HMBC spectra was performed to confirm these results. The HMBC correlations from the hydroxyl methylene protons (*δ*_H_ 4.73 and 4.56, H-30) to C-20 (*δ*_C_ 156.5), C-29 (*δ*_C_ 107.3), and C-19 (*δ*_C_ 44.2) established the presence of a hydroxyl methylene group attached to the exocyclic double bond located at the pentacyclic E ring (Figure 2). Furthermore, the HMBC correlations from H-22 (*δ*_H_ 2.01 and 1.57)/H-16 (*δ*_H_ 1.93) to C-28 (*δ*_C_ 177.3), and from the methoxy group (*δ*_H_ 3.77) to C-28 (*δ*_C_ 177.3), indicated the position of the methyl ester carboxylic acid. The position of an additional carboxylic acid group in compound **1** was further determined by HMBC correlation from H-13 (*δ*_H_ 2.79)/H-15 (*δ*_H_ 1.57) to C-27 (*δ*_C_ 178.8). The relative configurations were determined by NOESY analysis. The NOESY correlations of H-3/H-23, H-3/H-9, and H-18*α*/H-19*α*, along with Me-24/Me-25, Me-25/Me-26, and Me-26/H-13*β*, were observed. Based on the well-established stereochemistry of lupane-type triterpenoids and their biosynthetic rules [30], the absolute configuration can be deduced. Therefore, the chemical structure of compound **1** was fully elucidated using ^1^H and ^13^C NMR, HSQC, HMBC, and NOESY spectroscopic techniques, and it was named cleistocalyxic acid L.

Compound **2** was obtained as a white amorphous powder, with its molecular formula determined as C_31_H_48_O_6_, based on HRESIMS data (*m*/*z* 515.3372 [M − H]^–^, calcd for 515.3373). The IR spectrum revealed absorption bands indicative of hydroxyl (3245 cm^–1^), ester (1718 and 1212 cm^–1^), and carboxyl (2934 and 1680 cm^–1^) groups. The ^1^H and ^13^C NMR spectra of compound **2** (Table 1) were similar to those of **1**, with the only difference being the change in the position of a hydroxyl group. The chemical shifts from *δ*_H_ 4.73 and 4.56 at C-30 in compound **1** had disappeared, and new chemical shifts at *δ*_H_ 4.10 and 3.59 at the C-23 in compound **2** appeared. Based on HSQC, two correlations from *δ*_H_ 4.10 and 3.59 (H-23) to C-23 (*δ*_C_ 68.5) suggested the presence of a hydroxyl group at C-23. HMBC correlations from H-23 to C-3 (*δ*_C_ 74.0), C-4 (*δ*_C_ 43.3), C-5 (*δ*_C_ 49.6), and C-24 (*δ*_C_ 13.5) clearly indicated the position of the hydroxyl group at C-23. Additionally, HMBC correlations from H-22 (*δ*_H_ 2.02 and 1.45)/H-18 (*δ*_H_ 2.13)/H-16 (*δ*_H_ 1.88) and the methoxy group (*δ*_H_ 3.77) to C-28 (*δ*_C_ 177.3) determined the position of methyl ester carboxylic acid. The HMBC correlations from H-13 (*δ*_H_ 2.71) and H-15 (*δ*_H_ 1.56) to C-27 (*δ*_C_ 178.8) established the position of carboxylic acid. In the NOESY spectrum, H-3/Me-23, H-3/H-5, H-5/H-9, Me-24/Me-25, Me-25/Me-26, Me-26/H-13, and H-13/H-19 were observed to determine the configuration of compound **2**. It was found to be consistent with that of compound **1** based on NOESY analysis. Similarly to compound **1**, the absolute configuration can be deduced based on biosynthetic rules. Therefore, the structure of compound **2** was determined and named cleistocalyxic acid M.

Compound **3**, a white amorphous powder, was determined to have a molecular formula of C_29_H_40_O_5_ by HRESIMS (*m*/*z* 513.2851 [M + HCOO]^−^, calcd for 513.2852). The IR spectrum indicated the presence of hydroxyl (3365 cm^−1^) and ester (1775 and 1149 cm^–1^) groups. The ^1^H NMR spectrum of compound **3** (Table 1) revealed two pairs of exomethylene protons [*δ*_H_ 5.89 and 4.99 (each 1H, br s, H-23), and *δ*_H_ 4.87 and 4.81 (each 1H, br s, H-30)], and four oxygenated protons [*δ*_H_ 4.38 (1H, d, *J* = 8.8 Hz, H-3), 4.13 (1H, m, H-2), 3.32 (1H, dd, *J* = 3.1, 2.4 Hz, H-11), and 3.07 (1H, d, *J* = 3.7 Hz, H-12)]. Three singlet methyl groups appeared at *δ*_H_ 1.18, 1.17, and 0.91, and a doublet methyl group was detected at *δ*_H_ 1.45 (3H, d, *J* = 6.3 Hz, H-29) in ^1^H NMR spectrum. The ^13^C NMR spectrum of compound **3** revealed 29 signals, indicating a nor-type triterpenoid. The ^13^C signals of compound **3** included a carboxylic group at *δ*_C_ 178.6, two exocyclic methylene groups at *δ*_C_ 152.4 (C-20) and 108.7 (C-30), *δ*_C_ 151.9 (C-4) and 105.7 (C-23), five oxygenated carbons at *δ*_C_ 89.7, 79.9, 73.7, 56.8, and 55.3, and four methyl groups at *δ*_C_ 20.7, 16.9, 16.7, and 16.3. The 1D NMR data of compound **3** were similar to those of ulmoidol isolated from *Ilex kudincha* [31]. The major difference between compound **3** and ulmoidol was the replacement of a methyl group at C-30 (*δ*_C_ 19.5) with a double bond [*δ*_C_ 151.9 (C-4) and 108.7 (C-30)]. The HMBC correlations from H-19 (*δ*_H_ 2.71) to C-29 (*δ*_C_ 16.9), C-30 (*δ*_C_ 108.7), C-18 (*δ*_C_ 61.6), and C-20 (*δ*_C_ 152.4) supported the presence of an ulmoidol moiety in structure. A proton signal at *δ*_H_ 1.88 (H-18) was extensively correlated with carbons in the HMBC spectrum, such as C-28 (*δ*_C_ 178.6), C-29 (*δ*_C_ 16.9), C-17 (*δ*_C_ 45.7), C-13 (*δ*_C_ 89.7), and C-12 (*δ*_C_ 56.8), indicating the presence of a lactone ring. Also, the HMBC correlation from Me-27 (*δ*_H_ 1.18) to C-13 (*δ*_C_ 89.7) supported the presence of a lactone moiety consisting of C-13, C-17, and C-28. In addition, the correlations from H-3 (*δ*_H_ 4.38) to C-2 (*δ*_C_ 73.7), C-4 (*δ*_C_ 151.9), and C-23 (*δ*_C_ 105.7) determined the position of the 23-nor-triterpenoid. Subsequent NOESY analysis confirmed the stereochemistry of the structure from the following correlations between H-3/H-5, H-5/H-9, H-9/Me-27, Me-27/H-19, H-2/Me-25, Me-25/Me-26, Me-25/H-11, H-11/H-12, Me-26/H-12, H-12/H-18, and H-18*β*/Me-29. Therefore, the structure of the new compound **3** was established and named cleistocalyxolide C.

The structures of 14 known compounds were determined by comparing their 1D and 2D NMR spectroscopic data with those reported in the literature. These compounds were identified as melaleucic acid 28-*O*-methyl ester (**4**) [29], betulinic acid (**5**), alphitolic acid (**6**) [32], ulmoidol (**7**) [31], (2a, 3b, 12a)-trihydorxy-olean-28-oic acid 28,13-lactone (**8**), 12-hydroxyasiatic acid (**9**), 2*α*,3*β*,19*α*-trihydroxy-24-norurs-4(23),12-dien-28-oic acid (**10**) [33], ilekudinol B (**11**) [31], oleanolic acid (**12**) [34], maslinic acid (**13**) [35], erythrodiol (**14**) [36], arjunolic acid (**15**) [37], 3-*O*-(*E*)-*p*-coumaroyl maslinic acid (**16**), and 3-*O*-(*Z*)-*p*-coumaroyl maslinic acid (**17**) [38] (Figure 1).

### 3.2. PTP1B Inhibitory Activity of Compounds ***1**−**17*** and Enzyme Kinetics

PTP1B has been established as an attractive drug target for treating diabetes [39,40,41]. To explore possible therapeutic potentials, we measured the inhibitory effects of compounds **1**–**17** on the PTP1B enzyme, with ursolic acid as a positive control (Appendix A). Firstly, we screened all isolated compounds at a concentration of 20 µM. Most isolates exhibited inhibitory activity against PTP1B except for cleistocalyxic acid L (**1**), cleistocalyxolide C (**3**), ulmoidol (**7**), and (2a,3b,12a)-trihydorxy-olean-28-oic acid 28,13-lactone (**8**). From this result, it can be inferred that the presence of a lactone ring in **3**, **7**, and **8** is associated with reduced PTP1B inhibitory activity. Following the initial screening, the half-maximal response of PTP1B activity (IC_50_) was determined for the compounds that showed less than 50% enzyme activity at a concentration of 20 μM. To determine the IC_50_ values, we employed six different concentrations for each compound to establish their inhibitory potency profiles. The results revealed that betulinic acid (**5**), oleanolic acid (**12**), maslinic acid (**13**), and 3-*O*-(*Z*)-*p*-coumaroyl maslinic acid (**17**) exhibited significant inhibitory activity on the PTP1B enzyme, similar to the positive control (Table 2 and Appendix A). The PTP1B inhibitory effects of betulinic acid, oleanolic acid, and maslinic acid have been previously reported, and our results support these previous reports.

We selected one compound each from different triterpenoid backbones, including lupane, ursane, and oleanane types, for further study. The enzymatic inhibition mechanism of the selected compounds **6**, **9**,**13**, and **17** was deduced from experimental kinetic data, where different concentrations of the *p*-NPP substrate and inhibitors were tested to measure the velocities of the PTP1B enzymatic reaction. The inverse of the velocity and the inverse of the substrate concentration values were used to create Lineweaver–Burk plots. These plots showed straight lines intersecting on the 1/[S] axis for compounds **6**, **9**, **13**, and **17** (Figure 3), indicating that these compounds act as non-competitive inhibitor of PTP1B. As a lupane-type triterpenoid, compound **6** shares the same skeletal structure with compounds **1**, **2**, **4**, and **5**. Compound **9** shows structural similarities with compounds **10** and **11**, while compounds **12**–**17** are oleanane-type triterpenoids. It is reasonable to assume that all the isolated compounds possess a non-competitive mechanism of action in inhibiting PTP1B since compounds **6**, **9**, **13**, and **17**, which are representative compounds of each type, showed non-competitive inhibition of PTP1B. We further examined Dixon plots, which depict the plot of 1/enzyme velocity (1/V) against inhibitor concentration (I), to determine the dissociation or inhibition constant (*K_i_*) for the enzyme–inhibitor complex. The value at which the x-axis intersects corresponds to—*K_i_*. The *K_i_* values for compounds **6**, **9**, **13**, and **17** were calculated to be 2.8 ± 0.4, 7.6 ± 1.2, 3.2 ± 0.6, and 2.7 ± 0.4 μM, respectively (Figure 3, Table 2). The *K_i_* value serves as an important indicator of the binding affinity between an enzyme and an inhibitor. For non-competitive inhibitors, the IC_50_ is expected to equal the *K_i_*. Our experimental data show that the values obtained for both parameters are consistent (Table 2).

### 3.3. Structure Activity Relationship (SAR) Analysis and Molecular Docking Experiments

The isolated compounds predominantly feature a pentacyclic triterpenoid ring, enabling us to discern the impact of various functional groups within this structural framework on PTP1B inhibition. Compounds **3**, **7**, and **8** exhibit reduced inhibitory effects, likely due to the presence of a lactone ring connecting C-13 and C-18. Similarly, hydroxylation of methyl groups at Me-24 and Me-30 leads to diminished activity among lupane-type triterpenes, including compounds **1**, **2**, and **4**–**6**. Adding functional groups to the ursolic acid skeleton, such as the C-3 and C-19 hydroxyl groups in compounds **9** and **10**, results in a slight reduction in PTP1B inhibition. Additionally, a decrease in activity is observed with methyl modifications at C-24 and C-25 in compounds **10** and **11**, leading to a more noticeable decline in bioactivity compared to compound **9** and ursolic acid. The inhibition efficacy of oleanolic acid (**12**) is slightly improved by incorporating hydroxyl groups at position C-2. Furthermore, substituting the carboxylic acid at C-28 with a hydroxyl group diminishes its activity. Adding a *p*-coumaroyl moiety to maslinic acid (13) results in a slight decrease in activity for compounds **16** and **17**.

Molecular docking simulations were employed to better understand the chemical interactions between the isolated compounds and the PTP1B protein. The selection of the 1T49 crystal structure for molecular docking studies was based on prior published research and the presence of a crystallized ligand at the allosteric site [27,42,43]. The docking results yielded various binding poses for the isolated molecules (Table 3). A detailed evaluation was performed, and the pose with the most favorable score was chosen for further examination. Most molecules exhibited binding poses that fit well into the hydrophobic pocket of the allosteric site (Appendix A). In the case of lupane derivatives, key interactions were observed between the olefinic carbons and the amino acids leucine-192 and phenylalanine-196, resulting in a π-alkyl interaction. Hydroxylation at C-30 may hinder this interaction, as seen in compound **1**. Alkyl interactions were also noted between Me-25 and the backbone of lysine-197. Moreover, conventional hydrogen bond interactions were confirmed between the carboxylic acid and alanine-193 and lysine-197. Substituting it with a methyl ester carboxylic acid appears to eliminate this interaction. The additional hydroxyl group at position Me-24 in ursane-type triterpenoids may facilitate other interactions, such as conventional hydrogen bonds, as evidenced by the interaction between this hydroxyl group and glutamate-276 (Figure 4). Additional hydrogen bond interactions between the carboxylic acid and alanine-193 and lysine-197 were observed for ursane- and oleanane-type triterpenoids. Similarly, compound **13** showed hydrophobic interactions with Phe280 and Ala 189. Also, there were hydrogen bond interactions between the carboxylic acid group and Lys197. The inclusion of a coumaroyl substituent at position C-3, as seen in compound **16** and **17**, resulted in binding poses with additional π-π interactions between the aromatic ring in the coumaroyl moiety and phenylalanine-280. These poses showed better docking scores than compounds **13**. However, this did not translate into better performance in the observed experimental enzymatic inhibitory activity.

In conclusion, preserving an unmodified hydrophobic core is crucial for maintaining inhibitory activity in PTP1B. The incorporation of polar groups at the core, particularly in the C and D rings, results in a reduction in binding affinity. A carboxylic acid moiety at C-28 is also pivotal in facilitating hydrogen bonding with polar protein residues. Substituents at C-3, as demonstrated by compounds **16** and **17**, did not significantly affect enzymatic activity and can contribute to enhanced binding affinity. These insights into the structural determinants of PTP1B inhibition provide valuable information for developing future derivatives based on isolated triterpene scaffolds.

### 3.4. Effect of Stimulating Glucose Uptake in Adipocytes

To determine if the PTP1B inhibition activity of compounds **1**–**17** affects the glucose uptake process, we also further evaluated their effects in vitro using 2-NBDG in 3T3-L1 adipocyte cells. 2-NBDG is a known fluorescent-tagged glucose probe used for identifying insulin-mimetic compounds [44,45]. The 3T3-L1 adipocytes, insulin-sensitive cells that are fully differentiated, were utilized for this experiment. To determine the transportation efficacy of 2-NBDG into the cells, we assessed the fluorescent signal in the differentiated adipocytes after treating them with each isolate at a concentration of 20 μM using fluorescence microscopy, with insulin serving as a positive control (Figure 5). As a result, compounds **6**, **12**, **13**, **16**, and **17**, which also exhibited PTP1B inhibitory activities, showed glucose uptake stimulatory effects. Among them, compounds **6**, **13** and **17** displayed particularly potent glucose uptake stimulatory effects, prompting further investigation for these compounds at various concentrations (5, 10, and 20 μM). The activities of compounds **6**, **13**, and **17** on glucose uptake were found to be dose dependent (Figure 5).

## 4. Discussion

PTP1B is a promising target for diabetes control due to its role in the development of insulin resistance, where it plays a key role in regulating insulin signaling. Triterpenes have been proven to possess anti-diabetic potential and have inhibitory effects on PTP1B [16,46,47]. Several studies have highlighted the ability of triterpenoids to improve glucose uptake [44,45]. Oleanolic acid (**12**) has been shown to improve glucose homeostasis and prevent the progression of type 2 diabetes in pre-diabetic male Sprague Dawley rats [44]. Furthermore, maslinic acid (**13**), a triterpenoid isolated in this study, increased glucose uptake, reduced lipid droplet and triglyceride levels, and raised intracellular Ca^2+^ concentration, suggesting its potential as a candidate for obesity and diabetes treatment [48]. Among the triterpenes in *C. operculatus* leaves, maslinic acid, corosolic acid, asiatic acid, and arjulonic acid are the most abundant [24]. The content of total terpenoids in hot-water infusion from *C. operculatus* leaves has been reported as 13.12 mg/g of dried leaves [49]. Thus, it is natural to assume that *C. operculatus* leaves may exert anti-diabetic properties, but no prior research has reported it.

In this study, the isolation of 17 triterpenes with various substitution patterns facilitated the elucidation of their structure–activity relationships concerning glucose uptake efficacy and PTP1B inhibition. Interestingly, we found that compounds inhibiting PTP1B also stimulated glucose uptake. A moderate positive correlation was observed between glucose uptake activity and PTP1B inhibition (Spearman correlation, *R* = 0.51, *p* = 0.03; see Appendix A). This suggests that changes affecting PTP1B inhibitory activity will likely influence glucose uptake stimulation. Therefore, the stimulatory effect on glucose uptake appears to be mediated through PTP1B. It is well established that the dephosphorylation of insulin receptor substrates (IRS) by PTP1B affects the translocation of glucose transporter type 4 (GLUT4), significantly reducing glucose uptake in adipose and skeletal muscle cells [50,51,52,53,54,55]. This known mechanism supports the correlation observed in our experimental results.

Betulinic acid (**5**) showed slightly better activity than alphitolic acid (**6**) in inhibiting PTP1B. However, the additional hydroxyl group at C-2 significantly improved glucose uptake activity of **6**, likely due to enhanced solubility or transport through cell membranes (Appendix A). Similarly, the hydroxyl group at C-2 improved the activity of maslinic acid (**13**) compared to oleanolic acid (**12**). Hydroxylation at other positions compromises PTP1B inhibition and glucose uptake stimulation. On the other hand, adding a *p*-coumaroyl moiety to maslinic acid (**13**) did not significantly impact either PTP1B inhibitory activity or glucose uptake. Modifications of ursolic acid skeleton reduced PTP1B inhibition for compounds **9**–**11**. Compounds **3**, **7**, and **8** exhibited low glucose uptake stimulation and PTP1B inhibition activity, likely due to a lactone ring connecting C-13 and C-18. Despite the known bioactivities of betulinic acid (**5**), oleanolic acid (**12**), maslinic acid (**13**), and ursolic acid on PTP1B, we further confirmed their effects on glucose uptake stimulation. In addition, we observed that, in general, compounds retained the same levels of bioactivity. However, alphitolic acid (**6**), maslinic acid (**13**), and 3-O-(Z)-*p*-coumaroyl maslinic acid (**17**) outperformed the others in the glucose uptake assay in 3T3-L1 adipocytes, possibly due to better solubility and the ability to absorb through cell membranes.

Overall, we anticipate that the findings of this study will contribute significantly to the development of standardized extracts of *C. operculatus* as a promising option for complementary diabetes treatment.

## 5. Conclusions

During our investigation of *Cleistocalyx operculatus* leaves, we isolated three new triterpenoids (**1**−**3**) and 14 known compounds (**4**−**17**). All these compounds were tested in PTP1B inhibition assays, and their IC_50_ values were determined. The most active compounds were betulinic acid (**5**), oleanolic acid (**12**), and maslinic acid (**13**). Enzyme kinetic experiments revealed that compounds possess a non-competitive inhibition mechanism, which molecular docking models supported. In vitro glucose uptake assays in differentiated 3T3-L1 adipocytes showed that alphitolic acid (**6)**, maslinic acid (**13)**, and 3-O-(Z)-*p*-coumaroyl maslinic acid (**17**) exhibited the most potent anti-diabetic effects. In addition, we infer that the glucose uptake stimulation effect by the isolated compounds is mediated through PTP1B, as indicated by the moderate positive correlation. Overall, this work provides experimental evidence supporting the potential of *C. operculatus* leaf extract as a plant-based alternative for diabetes management.

## Figures and Tables

**Figure 1 nutrients-16-02839-f001:**
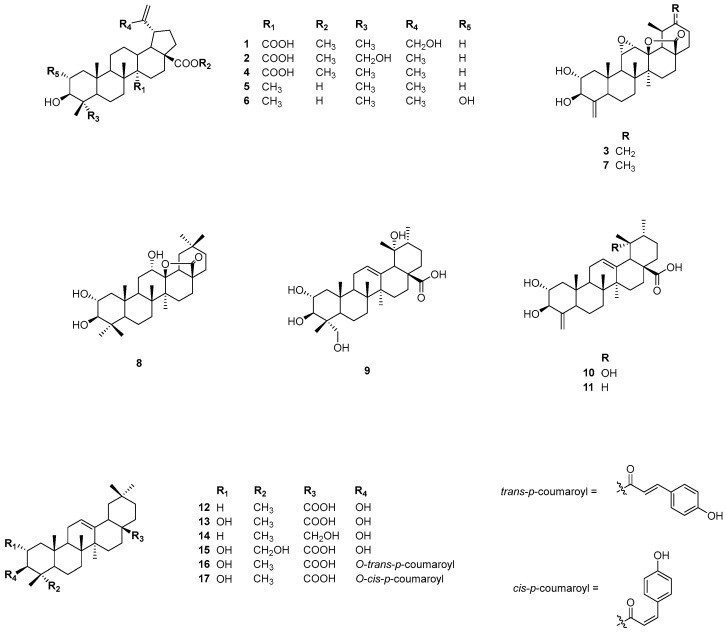
Chemical structures of isolated compounds **1**−**17** from the leaves of *C. operculatus*.

**Figure 2 nutrients-16-02839-f002:**
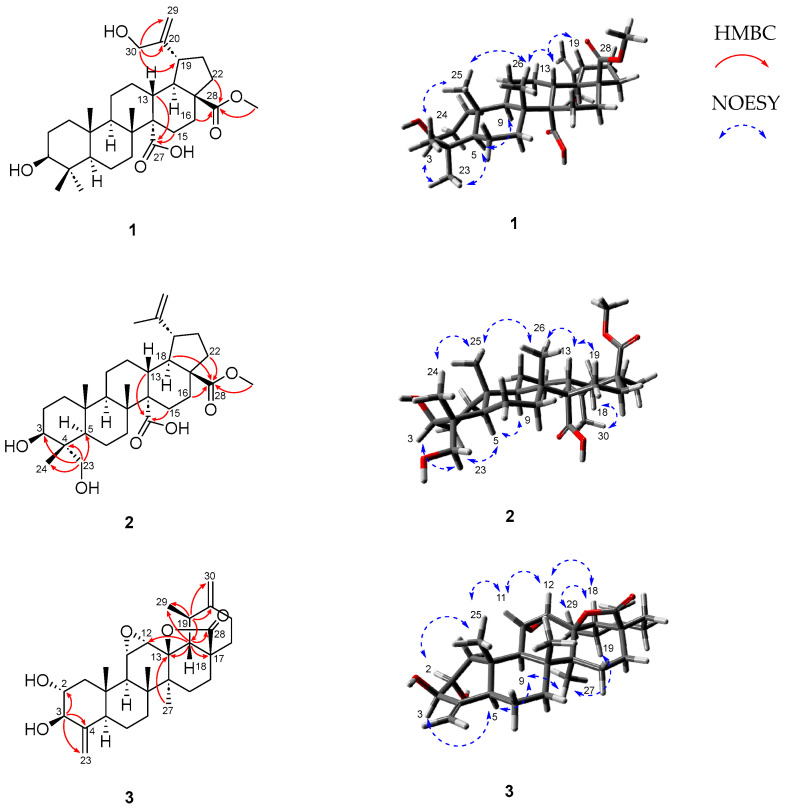
Key HMBC correlations (red) and NOESY correlations (blue) of **1**−**3**.

**Figure 3 nutrients-16-02839-f003:**
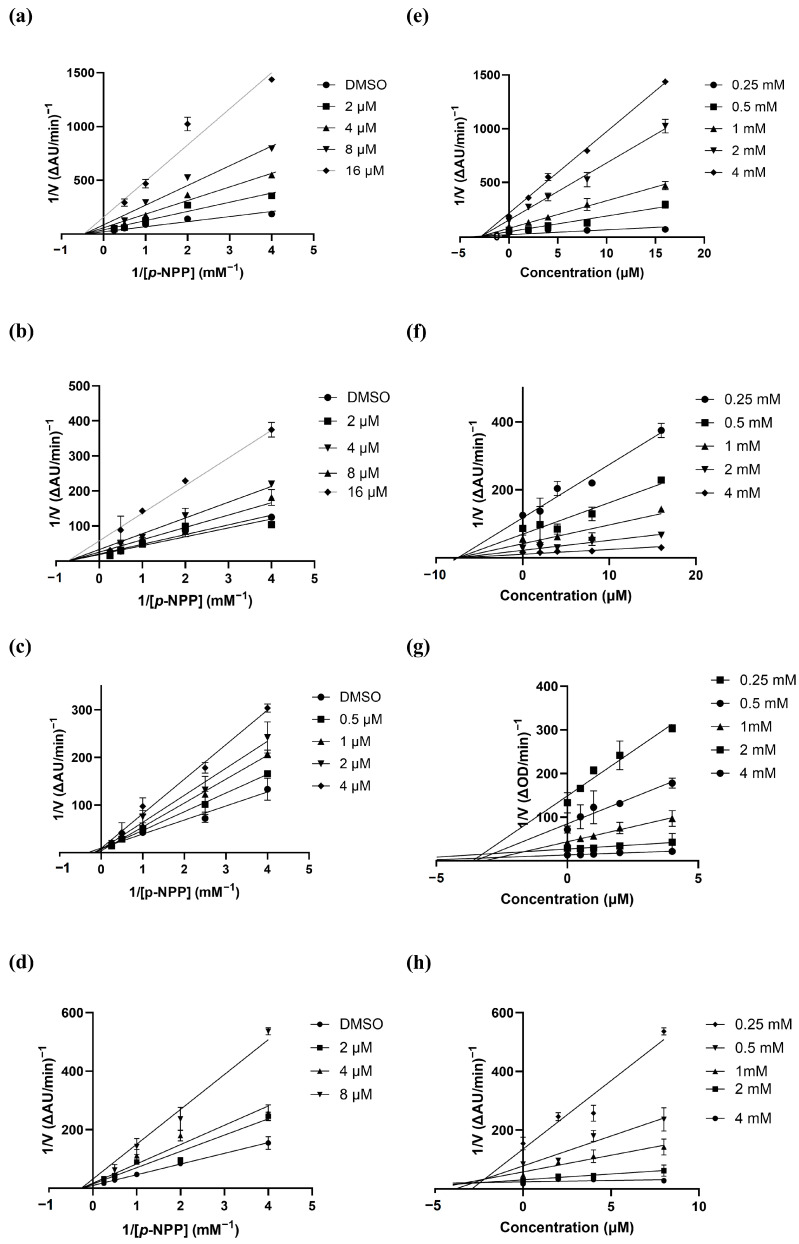
Lineweaver–Burk plots for the inhibition of PTP1B enzyme by **6**, **9**, **13**, and **17**; (**a**), (**b**), (**c**), and (**d**), respectively. Dixon plots for **6**, **9**, **13**, and **17**; (**e**), (**f**), (**g**), and (**h**), respectively.

**Figure 4 nutrients-16-02839-f004:**
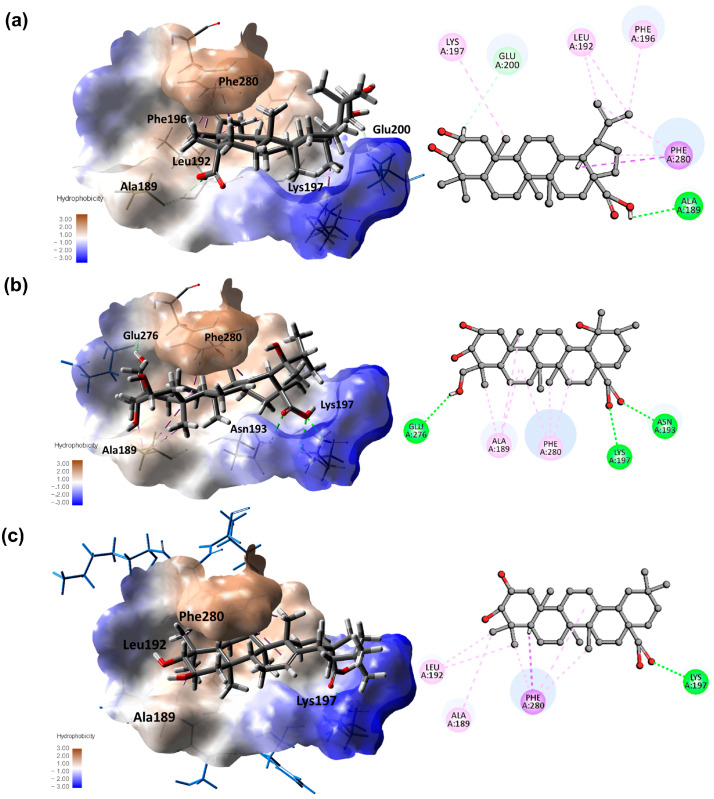
Molecular docking models for PTP1B (PDB:14T9) inhibition at allosteric site for compounds **6** ((**a**), 96.83 Libdock score), **9** ((**b**), 81.57 Libdock score), **13** ((**c**), 81.57 Libdock score), and **17** ((**d**), 104.44, 106.92 Libdock score). Only hydrogens involved in interactions are shown in the 2D diagram.

**Figure 5 nutrients-16-02839-f005:**
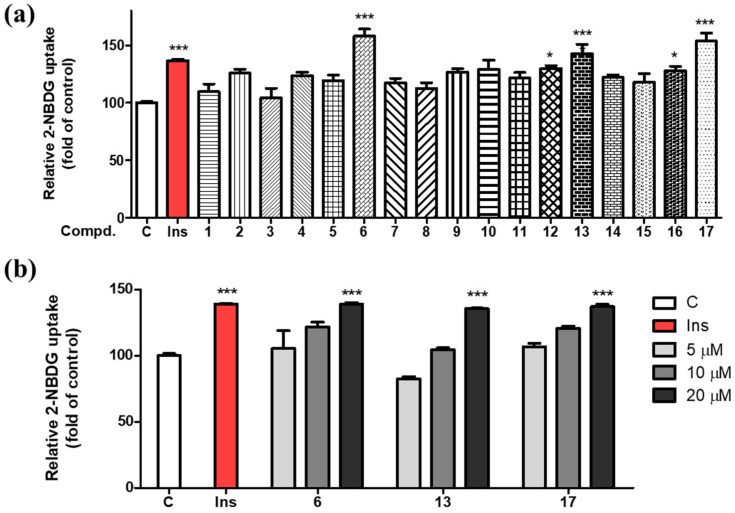
Stimulation effects of compounds **1**–**17** on glucose uptake in 3T3-L1 adipocytes using a fluorescent analogue of glucose, 2-NBDG. (**a**) 3T3-L1 adipocytes were incubated with insulin (100 nM) and compounds **1**–**17** (20 μM) for 1 h. The results are presented as the means ± S.D. (*n* = 3); each experiment was performed in triplicate; * *p* < 0.05, and *** *p* < 0.001, compared to the negative control. (**b**) Concentration−response relationships of the effects of compounds **6**, **13**, and **17** on glucose uptake in 3T3-L1 adipocytes. The cells were treated with these compounds at 5, 10, and 20 μM or 100 nM insulin concentrations. After 1 h of incubation, fluorescence intensities were measured using fluorescence microscopy.

**Table 1 nutrients-16-02839-t001:** ^1^H (600 MHz) and ^13^C NMR (150 MHz) spectroscopic data for compounds **1**–**3**.

Position	1		2		3	
	*δ*_H_ (*J* in Hz)	*δ* _C_	*δ*_H_ (*J* in Hz)	*δ* _C_	*δ*_H_ (*J* in Hz)	*δ* _C_
**1**	1.73 d (13.1) 0.99 m	39.8	1.75 m 1.08 m	39.6	2.62 dd (12.7, 5.0) 1.72 m	48.2
**2**	1.86 m 1.80 m	28.8	1.92 m 1.86 m	28.3	4.13 m	73.7
**3**	3.27 dd, (11.3, 4.7)	78.3	4.05 dd (11.3, 4.0)	74.0	4.38 d (8.8)	79.9
**4**		39.9		43.3		151.9
**5**	0.88 m	56.5	1.54 m	49.6	5.04 br d (12.2)	50.4
**6**	1.62 m 1.42 m	19.3	1.68 m 1.45 m	19.2	1.64 m 1.50 m	21.1
**7**	3.01 m 1.93 m	38.8	2.15 m 1.87 m	38.6	1.28 m 1.06 m	30.5
**8**		41.3		41.3		42.3
**9**	1.93 m	52.2	2.03 m	52.2	1.90 br s	50.1
**10**		38.3		38.2		38.7
**11**	1.57 m 1.31 m	21.7	1.60 m 1.35 m	21.7	3.32 dd (3.1, 2.4)	55.3
**12**	2.79 m 1.89 m	28.1	2.70 m 2.01 m	27.4	3.07 d (3.7)	56.8
**13**	2.79 m	40.7	2.71 m	40.7		89.7
**14**		60.3		60.4		42.2
**15**	2.50 d (13.4) 1.57 m	29.1	2.47 d (13.4) 1.56 m	29.0	1.73 m 1.03 m	27.5
**16**	2.68 m 1.93 m	35.3	2.65 m 1.88 m	35.4	2.31 m 1.42 m	23.5
**17**		57.3		57.3		45.7
**18**	2.33 t (11.1)	53.0	2.13 m	52.9	1.88 m	61.6
**19**	3.53 td (11.3, 4.7)	44.2	3.51 td (10.9, 4.2)	48.2	2.71 m	36.6
**20**		156.5		151.2		152.4
**21**	2.20 m 1.63 m	33.0	2.03 m 1.46 m	31.3	2.23 m 2.29 m	32.4
**22**	2.01 m 1.57 m	37.5	2.02 m 1.45 m	37.7	1.96 dt (12.9, 3.1) 1.62 m	33.8
**23**	1.08 s	29.0	4.10 d (10.3) 3.59 d (10.3)	68.5	5.89 br s 4.99 br s	105.7
**24**	1.01 s	17.0	1.06 s	13.5		
**25**	0.91 s	17.5	0.99 s	17.8	0.91 s	16.3
**26**	1.18 s	17.9	1.20 s	17.9	1.17 s	20.7
**27**		178.8		178.8	1.18 s	16.7
**28**		177.3		177.4		178.6
**29**	5.61 br s 5.38 br s	107.3	5.04 br s 4.81 br s	111.0	1.45 d (6.3)	16.9
**30**	4.73 d (15.0) 4.56 d (15.0)	64.7	1.89 s	19.7	4.87 br s 4.81 br s	108.7
OCH_3_	3.77 s	51.9	3.77 s	52.0		

**Table 2 nutrients-16-02839-t002:** Inhibitory activity of isolated compounds against PTP1B enzyme ^a^.

Compounds	IC_50_ (μM)	Inhibition Type ^b^	*K_i_* (μM) ^c^
**1**	>20	-	-
**2**	>20	-	-
**3**	>20	-	-
**4**	8.7 ± 0.6	-	-
**5**	2.2 ± 0.1	-	-
**6**	4.1 ± 0.2	Non-competitive	2.8 ± 0.4
**7**	>20	-	-
**8**	>20	-	-
**9**	4.4 ± 0.5	Non-competitive	7.6 ± 1.2
**10**	12.2 ± 0.3	-	-
**11**	14.1 ± 0.3	-	-
**12**	1.7 ± 0.8	-	-
**13**	1.28 ± 0.1	Non-competitive	3.2 ± 0.6
**14**	3.6 ± 0.2	-	-
**15**	6.6 ± 0.6	-	-
**16**	3.6 ± 0.4	-	-
**17**	2.6 ± 0.3	Non-competitive	2.7 ± 0.4
Ursolic acid ^d^	2.9 ± 0.2 µM	-	-

(-) No test. ^a^ The values (μM) indicate 50% PTP1B inhibitory effects. These data are expressed as the mean ± S.E.M. of triplicate experiments. ^b^ Determined by Lineweaver–Burk plots. ^c^ Determined by Dixon plot interpretation. ^d^ Positive control.

**Table 3 nutrients-16-02839-t003:** Interacting residues and docking scores of **6**, **9**, **13**, and **17** in PTP1B obtained using Libdock Protocol in Discovery Studio.

	Libdock Score	Hydrogen Bond	Hydrophobic			Other Interactions	
			π-π Stacked	π-σ	π-Alkyl	π Anion/Cation	π-Sulfur
**6**	96.83	Ala189 Glu200		Phe280	Lys197 Leu192 Phe196		
**9**	81.57	Asn193 Lys197 Glu276			Ala189 Phe280		
**13**	104.44	Lys197		Phe 280	Ala189 Leu192		
**17**	106.92	Lys197 Lys279		Phe280	Ala189 Leu192		
Docking control (PDB:1T4J ligand)	148.67	Asn193 Lys197 Phe280	Phe280		Ala189 Leu192 Phe196	Phe280	Met182

## Data Availability

The data in this study are available by contacting the corresponding author.

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
