# Peer review of "In Vitro and In Silico Analysis of PTP1B Inhibitors from Cleistocalyx operculatus Leaves and Their Effect on Glucose Uptake"

_nutrients, 2024, doi:10.3390/nu16172839_

Round 1

Reviewer 1 Report

Comments and Suggestions for Authors

Won et al. submitted the manuscript entitled: In Vitro and In Silico Analysis of PTP1B Inhibitors from Cleistocalyx operculatus Leaves and Their Impact on Glucose Uptake, in which they reported isolation and structural elucidation of 17 pentacyclic triterpenes, as well as tested their PTP1B inhibitory activity and efficacy on glucose uptake. The authors also performed molecular docking to predict the binding poses with PTP1B. Generally, this is a well prepared manuscript and the topic would be of interest to potential readers of Nutrients.

I have some minor comments as follows.

1. The authors seemed to upload a wrong version of SI.

2. Page 13, figure 4: Did the authors ionized these ligands in ligand preparation step? Seems in figure 4a, the acid H still present but in figure 4b/c not. It’s weird because carboxylic acid hydrogen should be ionized after ligand preparation. Besides, the polarity H should be visible even they do not contribute to any interactions.

3. Page 14, figure 5: a) It is recommended to include kinetic assay data of compound 13 for consistency. b) A discussion should be involved on why data in glucose uptake assay showed higher value compared with that in PTP1B enzymatic assay.

Author Response

Won et al. submitted the manuscript entitled: In Vitro and In Silico Analysis of PTP1B Inhibitors from Cleistocalyx operculatus Leaves and Their Impact on Glucose Uptake, in which they reported isolation and structural elucidation of 17 pentacyclic triterpenes, as well as tested their PTP1B inhibitory activity and efficacy on glucose uptake. The authors also performed molecular docking to predict the binding poses with PTP1B. Generally, this is a well prepared manuscript and the topic would be of interest to potential readers of Nutrients.

I have some minor comments as follows.

à Response: We appreciate the reviewer’s positive feedback on our manuscript entitled "In Vitro and In Silico Analysis of PTP1B Inhibitors from Cleistocalyx operculatus Leaves and Their Impact on Glucose Uptake." We are pleased that the reviewer found the manuscript to be well-prepared and of interest to potential readers of Nutrients. We have carefully considered the reviewer’s comments and suggestions, and we believe that the revisions have strengthened the manuscript further. Thank you for your thoughtful review and support of our work.

  1. The authors seemed to upload a wrong version of SI.

à Response: Thank you for your critical comments. We will ensure that the most recent version of the Supporting Information is uploaded this time. We apologize for the inconvenience caused by the upload issues with the Supporting Information.

  1. Page 13, figure 4: Did the authors ionized these ligands in ligand preparation step? Seems in figure 4a, the acid H still present but in figure 4b/c not. It’s weird because carboxylic acid hydrogen should be ionized after ligand preparation. Besides, the polarity H should be visible even they do not contribute to any interactions.

à Response: Thank you for your expert and very thorough review. We appreciate the opportunity to clarify the methodology used in our study. During the ligand preparation protocol on Discovery Studio (Biovia 2024), we utilized both ionized and non-ionized forms of the molecules for docking experiments. After executing the Libdock docking protocol, we filtered the poses based on the best docking scores and subsequently analyzed the highest-scoring pose for each compound. Interestingly, in all cases, the ionized forms of the molecules didn’t yield high docking score. Therefore, the structures depicted in Figure 4 are all non-ionized forms.

However, it is important to note that the Discovery studio visualizer doesn’t display hydrogens that are not involved in interactions within the 2D diagram, even if they are polar hydrogens. This is why, in Figure 4a, the carboxylic acid hydrogen, which participates in hydrogen bonding as a donor, is visible. In contrast, in Figures 4b and 4 c, this hydrogen is not shown because it does not participate in any interaction. To provide further clarity, we have included a screenshot of compounds 6 and 9 as they appear in the Discovery studio software. Despite this, we have added a comment in the figure legend to explain this particular issue and ensure that it does not lead to any confusion. We hope this explanation provides the necessary clarification and look forward to any further feedback you may have..

  1. Page 14, figure 5: a) It is recommended to include kinetic assay data of compound 13 for consistency. b) A discussion should be involved on why data in glucose uptake assay showed higher value compared with that in PTP1B enzymatic assay.

à Response: Thank you for your excellent comment. We have taken your suggestion into account and have included the data for compound 13 in the revised manuscript. Additionally, we have expanded on the discussion regarding the results of PTP1B inhibition and its connection to the glucose uptake assay. To summarize our finding, we propose that for compounds to effectively inhibit PTP1B in cellular environments and subsequently increase glucose uptake, they must possess the ability to permeate cell membranes. Therefore, maybe compounds such as 6, 13, and 17, which are predicted to exhibit better absorption properties, may demonstrate enhanced glucose uptake activity when compared to other structurally similar compounds, potentially even outperforming those that exhibit stronger PTP1B inhibition in enzymatic assays.

Reviewer 2 Report

Comments and Suggestions for Authors

Review Report for Nutrients- 3162589

Jorge-Eduardo et al. conducted a chemical investigation of Cleistocalyx operculatus from Vietnam. Three new triterpenoids, two previously lupane-type triterpenoids and one nor-taraxastane-type triterpenoid, along with 14 known triterpenoids were identified. These compounds were comprehensively elucidated using HRMS and NMR, with UV and optical rotation measurements also included. In addition, their inhibitory activity towards PTP1B was evaluated. Several compounds exhibiting moderate inhibitory activity, with the IC50 values as low as 1.28 uM. The inhibitory mechanism for selected compounds was also determined. Further glucose uptake assay clearly indicates that compounds 6, 13, and 17 exhibited strong glucose absorption stimulation activity.

I believe this manuscript fits well within the readership of Nutrients. Here are a few issues that need to be addressed prior to acceptance for publication:

1.       In line 57-58, the aqueous extract has been reported to decrease glucose. It is highly recommended to claim the reason for researching 50% ethanol extract in this manuscript. Any similar contents found?

2.       Regarding the structural elucidation, (1) Key COSY signals should be labeled (as bold bonds); (2) The legend for the arrows, HMBC and NOSEY should be included in figure 2.

The absolute configuration of compound 1-3 should be determined.

3.       Line 448, PTP1B, all should be capitalized.

4.       It looks like compound 5, 12, 13 has the best PTP1B inhibitory potency. Need to explain why choose 6, 9, 17 with medium PTP IC50 for the kinetic analysis.

5.       It is recommended to determined the selectivity toward PTP1B over related subtypes, for example, highly conserved PTPN2.

6.       In Figure 3, The specific compound used should be included in each diagram. 3d) need to be revised. Move the Label 3e) to the top of the diagram.

7.       The predicted ADMET parameters of optimal compounds are recommended to be included.

Comments on the Quality of English Language

No comments

Author Response

Jorge-Eduardo et al. conducted a chemical investigation of Cleistocalyx operculatus from Vietnam. Three new triterpenoids, two previously lupane-type triterpenoids and one nor-taraxastane-type triterpenoid, along with 14 known triterpenoids were identified. These compounds were comprehensively elucidated using HRMS and NMR, with UV and optical rotation measurements also included. In addition, their inhibitory activity towards PTP1B was evaluated. Several compounds exhibiting moderate inhibitory activity, with the IC50 values as low as 1.28 uM. The inhibitory mechanism for selected compounds was also determined. Further glucose uptake assay clearly indicates that compounds 6, 13, and 17 exhibited strong glucose absorption stimulation activity.

I believe this manuscript fits well within the readership of Nutrients. Here are a few issues that need to be addressed prior to acceptance for publication:

à Response: Thank you for your positive comments and for recognizing the relevance of our work to the readership of Nutrients. We are pleased that you found our research on Cleistocalyx operculatus and the identified triterpenoids to be a valuable contribution. We appreciate your constructive comments and will address the issues you’ve highlighted to ensure that the manuscript meets the highest standards before publication. Thank you again for your support and guidance throughout this process.

  1. In line 57-58, the aqueous extract has been reported to decrease glucose. It is highly recommended to claim the reason for researching 50% ethanol extract in this manuscript. Any similar contents found?

à Response: Thank you for your excellent observation and very important question. We used the 50% ethanol extract to compare the full range of compounds present in Cleistocalyx operculatus (CO). We found that 50% ethanol provided a good balance between the yield of the extract and the polarity of the extracted compounds. Based on our experience, extracts with a higher water content mainly contain mainly primary metabolites and sugars, resulting in a lower yield of medium-polarity compounds. On the other hand, a higher content of organic solvent can increase the amount of chlorophylls. When we analyzed the contents of the 50% ethanol extract and hot water extract using UPLC qTOF MS/MS spectrometry, we confirmed that the 50% ethanol extract also included water-soluble metabolites too. To address the reviewer’s important question, we conducted the additional UPLC qTOF MS/MS spectrometry experiment and confirmed that the 50% ethanol extract contains compounds in the extract as well. We have included this information in the supplementary Information file (Supplementary Information Figure S32).

Figure S32. UPLC-qTOF MS/MS chromatogram of the hot water extract at 40 °C (upper chromatogram) and the 50% EtOH extract (lower chromatogram). The content of two extracts is similar in the polar to medium polarity region.

  1. Regarding the structural elucidation, (1) Key COSY signals should be labeled (as bold bonds); (2) The legend for the arrows, HMBC and NOSEY should be included in figure 2.

The absolute configuration of compound 1-3 should be determined.

à Response: Thank you for your excellent comment. We have included the legends in Figure 2. The relative configuration was determined based on NOESY correlations. Since the OH group at position 3 has a biosynthetically directed configuration in lupane and taraxastane triterpenoids, the absolute configuration was also established.

  1. Line 448, PTP1B, all should be capitalized.

à Response: Thank you for your careful inspection of the manuscript. All instances of “PTP1B” have been capitalized.

  1. It looks like compound 5, 12, 13 has the best PTP1B inhibitory potency. Need to explain why choose 6, 9, 17 with medium PTP IC50 for the kinetic analysis.

à Response: Thank you for your valuable comments. One compound of each class was selected based on the premise that similar skeletons may share a similar mechanism of inhibition. These compounds were chosen after confirming their activity on glucose uptake. For consistency between both assays, PTP1B inhibition and glucose uptake, the kinetic enzymatic data for the inhibition by compound 13 has now been included.

  1. It is recommended to determine the selectivity toward PTP1B over related subtypes, for example, highly conserved PTPN2.

à Response: Thank you for your valuable comments. Unfortunately, due to time constraints, we are unable to include this experiment in the current manuscript. However, if time permits and in future studies such as animal experiments or additional manuscript, we will consider your suggestion and proceed accordingly.

  1. In Figure 3, The specific compound used should be included in each diagram. 3d) need to be revised. Move the Label 3e) to the top of the diagram.

à Response: We appreciate your detailed review of our manuscript. We have made the necessary corrections as per your suggestions. The specific compound used has been included in each diagram, and the label for 3e) has been moved to the top of the diagram as requested.

  1. The predicted ADMET parameters of optimal compounds are recommended to be included.

à Response: Thanks for your recommendation. We have included the predicted ADMET parameters of the all isolated compounds in the Supporting Information file
